# Interoperable Test Cases to Mediate between Supply Chain's Test Processes

**Marco Franke [1,*] and Klaus-Dieter Thoben [1,2]**

1    Bremer Institut für Produktion und Logistik GmbH, 28359 Bremen, Germany
2    Faculty of Production Engineering, University of Bremen, 28359 Bremen, Germany
*    Correspondence: fma@biba.uni-bremen.de; Tel.: +49-04-2121-850089

**Abstract:** Heterogeneous test processes with respect to test script languages are an integral part of the development process of mechatronic systems that are carried out in supply chains. Up to now, test cases are not exchangeable between test processes because interoperability is not given. The developed approach enables the source-to-source compiling of test cases between test script languages. With this, the interoperability of test cases is achieved, and seamless integration within the supply chain is possible. The developed approach uses transcompilers as a baseline. In doing so, an interoperability model for test cases is presented. Based on the interoperability model, a source-to-source compiling for test cases is shown. The outcome is a prototype that handles test script languages, which are different with respect to type safety and applied programming paradigms. The approach ensures that test cases are still understandable and usable for test reports. The evaluation confirms the translation capabilities as well as the readability of the generated test case for the high-lift scenario from aviation. The interoperability of test cases within the supply chain enables the formalisation of procedural test knowledge to be used in a broad range of future scenarios, such as test automation, digital twins and predictive maintenance.

**Keywords:** machine-to-machine communications and interoperability; interoperability of test cases; case studies and tools; interoperability model for test cases; translation between test script languages





## 1. Introduction

The article aims to establish the interoperability of test cases from heterogeneous test script languages. Based on it, the exchange and the subsequent seamless integration of test cases within the supply chain's test processes of mechatronic systems are possible.

Mechatronics is a multidisciplinary engineering branch that enables mechatronic systems development. A mechatronic system is an intersection of the "Integration of electronics, control engineering, and mechanical engineering." as pointed out by [1]. Besides, [2] mentioned microprocessors' inclusion as part of the mechatronic system's electronics. The outcome of this engineering discipline is a compound of distributed, discrete-event systems [3]. Examples of mechatronic products, which are complex and safety-critical, are cars, trains, and aircraft. In the case of aircraft, the related verification effort represents a significant portion of the avionics development costs. For example, verifying software in avionics made up 40% of the development costs in 2001 [4]. The amount of software testing has been increasing [5] because of technological developments for integrating avionics subsystems [6]. The current rising trend is shown in Figure 1 [7].

Aircraft is developed using the V-model within a supply chain. Herby, the supply chains " . . . are more and more interdependent and interconnected for the purpose of collaboration, including vertical cooperation . . . " [8]. Within a supply chain's vertical cooperation, multiple test processes are heterogeneous regarding their goals, the applied test script languages and used test benches. A subset of the test cases from the supply chain has special characteristics. They are implemented by different suppliers and test the same

function under similar requirements but use different test script languages. Herby, the knowledge of how the mechatronic system is stimulated and checked is implemented in the test cases and is valuable. These test cases have similar semantics but different syntax. Suppliers could exchange them as resources. Supply chain management methods have evolved for that purpose and focus more on digitalisation. Current approaches related to digital twins [9,10], digital threads [11] and the enabling technologies for interoperability have a positive mediation effect on the supply chain [12]. Despite this mediation capability, the original test case is not interoperable yet cannot be integrated. This is because a test script language and a test bench are closely tightened, and this tuple is supplier specific. Thus, suppliers use different test script languages. The challenge is that the exchange of test cases is necessary in some cases. The manual extraction of the relevant information of the 'foreign' test case is not appropriate solution, because each test process is only familiar with its own test script language. Accordingly, the interoperability of test cases is necessary. The central research questions are how to represent the information inside a test case independently by test script languages and how to enable a source-to-source compiling of test cases. The solution enables the translation of a test case from the test script language into the target test script language and therefore, the seamless integration of 'foreign' test cases is possible.

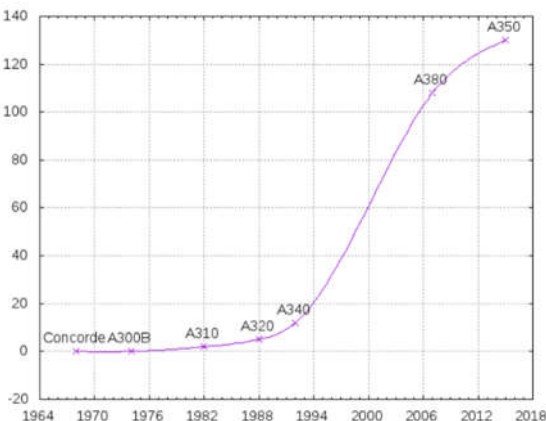

**Figure 1.** Airbus avionics software (MB), taken from [7].

With this, the approach handles test cases as data and translates them into information. Based on this, the typical approach of source-to-source compiling is applied, whereby an additional interoperability model is needed. An appropriate interoperability model could be modelled as ontologies known from Semantic Web, as established test models from model-based testing or state charts.

The remainder of this paper is structured as follows. Section 2 describes the conceptual foundations of this article and the related work. Afterwards, Section 3 presents the proposed approach, followed by the description of one application scenario in Section 4. After that, a short discussion is given. Finally, Section 6 presents a conclusion.

## 2. Background and Related Work

### 2.1. Hetergenous Test Processes in the Product Development

The first step of product development is the requirements analysis. This step is mandatory in all applied system development lifecycles like waterfall or V-Model. As part of the requirement analysis, standards such as ISO/IEC 12207:2017 [13] and ISO/IEC 15288:2008 [14] describe the declarative and procedural knowledge of the requirement analysis. With this, the process starts defining the mechatronic product's stakeholder requirements, which are the primary information source for design and testing. Subsequently, the stakeholder requirements are transformed into system and system elements requirements to define mechatronic systems' boundaries. Then, a supplier can perform the development process of a mechatronic system. For that purpose, the OEM's system

and system element requirements are the primary input source for the supplier-specific development process. The transformation of stakeholder requirements into system and system elements requirements is not executed just once by the OEM but rather recursively within the supply chain. The recursive translation mechanism, which is strongly simplified, is presented in Figure 2.

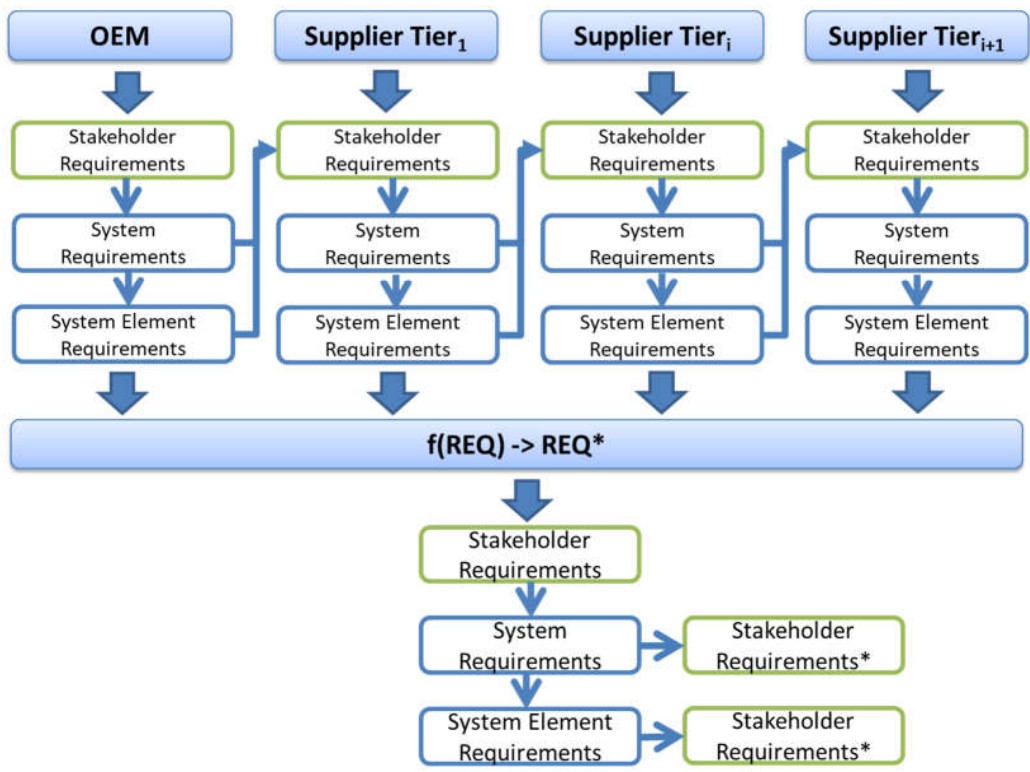

**Figure 2.** Transformation of requirements in the supply chain.

Figure 2 demonstrates the transformation of requirements within a network of entities whereby the entity type ranges from a department (internal source) to a supplier (external source). The transitive provision of requirement sets is hierarchical within the supply chain and starts at the OEM with stakeholder requirements describing the mechatronic product. The definition of a mechatronic system (as part of the mechatronic product) is based on a set of OEM's system- and system element requirements. This set represents a mechatronic product's function, independent and can consume all relevant inputs and deliver outputs via the interface. This requirement set is sent to a supplier as a requirement document for development purposes. With this, the role of the system requirements changes to stakeholder requirements at the receiver side (see Figure 3), which are used as the primary input for the test process and derive the system requirements again to specify the own system boundaries. As an outcome of the recursive transformation process, multiple requirements arise, interlinking between the suppliers.

The above-described recursive transformation of requirements directly impacts the test processes. Each test process for a mechatronic system checks the received stakeholder requirements and the derived system and system element requirements. Both types of requirements are necessary to enable a test process considering the provided function. In doing so, the system and system element requirements define the unit tests' scope, and the stakeholder requirements define the scope of integration tests. Figure 3 demonstrates the interplay of requirements types in more detail.

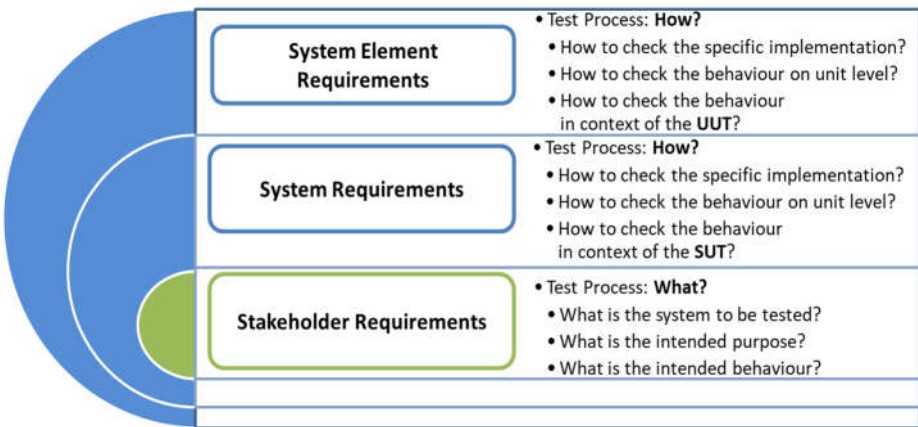

**Figure 3.** The application of requirement types in supply chain's test process.

The different system element requirements define the baseline of a test process, especially the architecture of the test bench and the test script language used. Although the stakeholder requirements can be the same, the system element requirements can differ significantly. Thus, the test processes in the supply chain are heterogeneous according to the test bench and test script language, while they share comparable stakeholder requirements.

The following example should clarify this. Let us take a system that moves the flap position of the aircraft at the push of a button. In the first test process, the correct functionality of the button (cabin test) is tested, and in the second test process, the movement of the flap (high lift test) is checked. While the test case in the first test process checks whether the button signal was transmitted after it was pressed, the test case in the second test process must continuously test whether the flap is moving correctly and, in an emergency, stop the test case within a few milliseconds to avoid damage to the shaft. Both test cases require the ability to read and write electronic control signals. The difference between the two test cases is the runtime behaviour. While pressing the button and detecting the signal processing, it does not have any hard real-time requirements.

In contrast, the control and monitoring of the signals in the second test process must take place in hard real-time. As a result, the architecture of the test bench in the second test process must meet real-time requirements, and the test script language feature monitor must also support the millisecond range. In contrast to the second test process, the first test process has no hard real-time requirements and has more options for the test bench and test script language selection. The above example makes it clear that there cannot be only one test script language and one test bench architecture.

The following describes whether the test processes' heterogeneity can negatively impact the development process.

### 2.2. Necessity and Added Value of Test Processes' Interoperability

Developing a complex product involves more than one supplier to design and test the mechatronic product. As pointed out previously, the supply chain's test processes focus on different goals and test levels. Therefore, they execute partially similar test cases from different test script languages and on different test benches. The following section presents the need to share test cases as information between test processes.

The test processes as part of the development process are shown in Figure 4. It is based on a supply chain and is structured as a supply pyramid. The primary goal of the lower layers of the pyramid is to deliver their products, which will be integrated and tested on the higher layers. In a faulty-free development process, a part is integrated into a component, a component is integrated into a system, and the system is integrated into an entire test bench to check the mechatronic product. In each of the test processes, the test cases of the supplier passed as well as the corresponding test cases from the integrator.

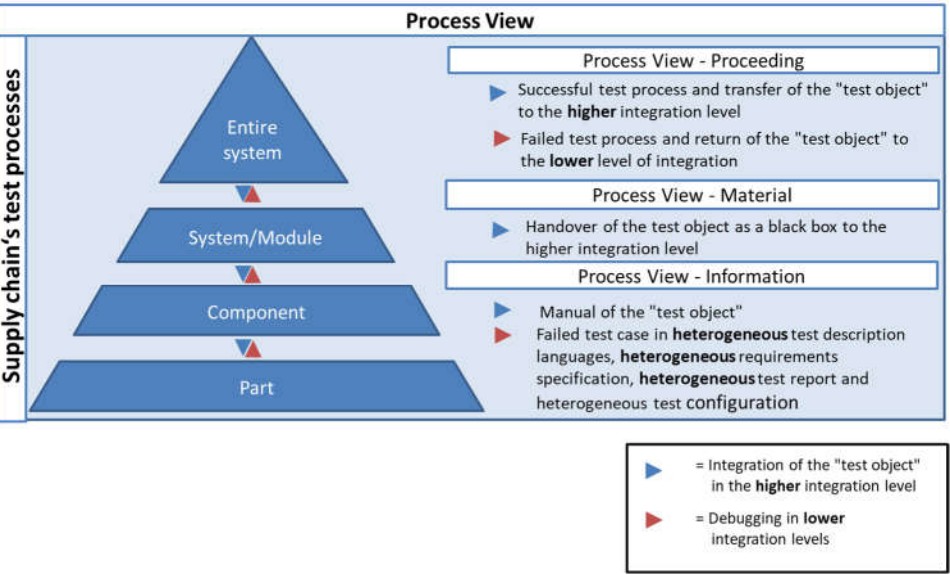

**Figure 4.** Collaboration of the supply chain's test processes.

However, this ideal sequence of test processes does not correspond to reality. There is a case in which the test process of the supplier pass and the subsequent test process of the integrator fails. In this situation, the failed test case is the foundation to start the failure search by the supplier.

The failure search involves the supplier and the integrator. The goal is to identify the root cause of the faulty test cases. A typical scenario for debugging as part of the failure search is to integrate the failed test case, including the integrator's configuration, into the test process of the supplier. Subsequently, the debugging tries to identify the following discrepancies that could be the root cause:

- Identify the difference in the configuration related to applied constants and signal definitions
- Identify different stimuli for achieving the same system behaviour in the UUT
- Identify different verification of the requirement-specific response of the UUT
- Identify different timing of the test cases executions related to different test benches

Identifying the discrepancies is partially possible with static analysis of the failed integrator's test case and the passed supplier test cases. The impact of the stimuli and the timing of the test case execution cannot be considered in a static analysis. To detect these types of discrepancies, the failed and passed test cases need to be executed on the supplier's test bench. For that purpose, the interoperability of test cases is required.

The challenge of the static analysis and the execution of the integrator's and supplier's test cases is that they are implemented in different test script languages as well as designed for different test benches. In the following, the term interoperability is defined and based on it, the relevant heterogeneity of test script languages is presented in detail.

*2.3. Interoperability as an Enabler for Information Exchange*

Interoperability is defined as "the ability of two or more systems or components to exchange and use information" [15]. The system of interest is the test bench, and exchangeable information is related to test cases. ISO 16100 has extended this definition of the term information as "the ability to share and exchange information using common syntax and semantics to meet an application-specific functional relationship through the use of a common interface" [16]. The syntax of established test script languages and the test bench as a common interface shall be the medium to exchange test cases within the supply chain's test processes. The semantics is the content of the test cases independent of the test script language's syntax. Similar semantics within test cases is possible in different test script languages if the test cases check the same function for the same product. The

semantics is a sequence of information on how the SUT shall be tested. In general, a piece of information is defined by Bellinger as " . . . data that has been given meaning by way of relational connection . . . " [17]. The data is the information represented in the syntax of a test script language.

The hierarchical taxonomy of data, information and knowledge is defined by the DIK pyramid [18]. Data transformation into the information needs an understanding of the test case statements. To achieve this understanding, the level of semantic understanding by Ören [19] needs to be applied, which contains the levels of Lexical understanding, Syntactical understanding, Morphological understanding and Semantic understanding. In doing so, the lexical, syntactical and morphological understanding enables the recognition of the structure of the information inside and the grouping of relevant entities in the test script language specific syntax. Semantic understanding is the crucial understanding of the data's meaning and closing the gap from a data to an information view. The syntax of the different test script languages is presented in the following to identify the needed capabilities for lexical, syntactical and morphological understanding. Moreover, the set of information which shall be extracted from the test cases is presented. Based on the set of information and the needed transformation capabilities, the interoperability of test cases can be developed.

### 2.4. Interoperable Test Cases as a Precondition for Test Processes' Interoperability

The interoperability of test cases is the capability to model the content of a test case as a procedural test knowledge independently of the selected test script language and to integrate its content for another test bench. In doing so, the integration means translating the content of the original test case into a test case for the target test script language. The interoperability of test cases is not possible yet, because test script languages are heterogenic. In the following, the heterogeneity of test script languages is presented in detail. With this, the shown heterogeneity needs to be harmonised to achieve interoperable test cases and enable an efficient failure search.

The available test script languages have so far been based on imperative programming. As a result, a test case consists of a set of statements executed sequentially. Although test cases have a similar structure, the test script languages are heterogeneous in terms of their syntax, semantics and provided functions. The following first describes the structure of a minimalist test case. Based on the minimalist test case, the reasons for the different implementations of the same test case structure in the test script languages are explained.

### 2.4.1. Structure of a Test Case

A test case consists of a sequence of statements. The sequence aims to prove that the SUT works according to the requirements. In the scope of testing information systems, it means that the specified test data, the expected results and a description of the steps to be tested with the expected results are recorded and mapped as a sequence of statements [20]. ISTQB and IEEE 829 give a congruent definition of the test case. According to this, a test case includes the following information: the preconditions necessary for the execution, the set of input values (one input value per parameter of the test object), the set of predicted results, and the expected postconditions. Test cases are developed concerning a specific goal or test condition, such as executing a specific program path or checking compliance with specific requirements [21].

Both definitions introduce the specified test data and the expected results as components of a test case. The test data is fed in various ways and stimulates the SUT. Based on the stimulation, results are expected on how the SUT has to behave according to the requirement. The expected results are defined as a condition, and the condition check is implemented using verdict type statements. Examples of verdicts are the statement expect in CCDL (https://www.razorcat.com/files/de/produkte/ccdl/Razorcat_Technical_Report_CCDL_Whitepaper_02.pdf accessed on 15 August 2022) or result in

RTT (https://www.dspace.com/de/gmb/home/products/sw/test_automation_software/automated_test_execution_in_re.cfm accessed on 15 August 2022).

A mechatronic product is a network of distributed, event-discrete systems [22], which is an information system that implements its control function using software [3]. Correspondingly, test cases for testing mechatronic systems contain statements for stimulating the SUT as well as verdicts. A mechatronic system as part of a mechatronic product behaves like a black box. Thus, it can only be stimulated via its interfaces. In addition, the conditions for the verdicts can only be defined via signals that are available via the interface. The integration of software and actuators means that the stimulation of a signal is not executed immediately but requires time, depending on the system. This time behaviour must be taken into account in the test cases. Statements for controlling the runtime behaviour are provided for this.

The structure of a minimalist test case, which contains the above-mentioned statements, was defined for testing mechatronic systems by [23], and a replica is shown in Figure 5.

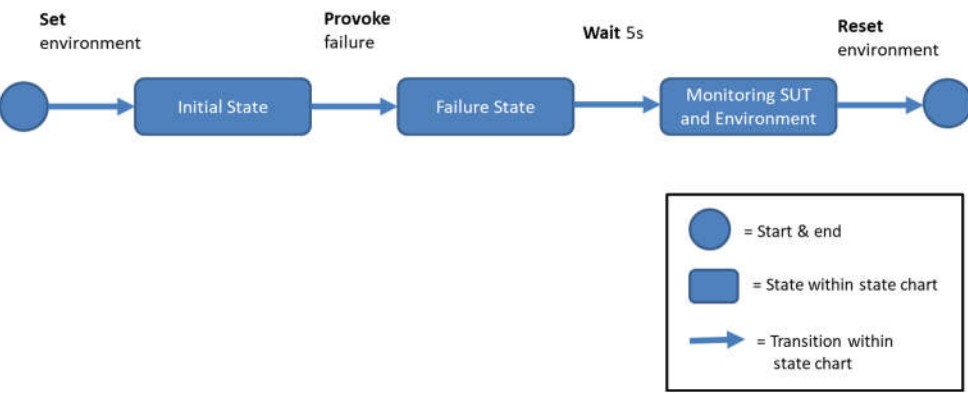

**Figure 5.** Basic steps of a test case.

As shown in Figure 5, a test case stimulates three consecutive states in the SUT. First, the test case stimulates the environment [24]. The error is then stimulated, and the SUT is finally observed. This proceeding of states is achieved by executing a sequence of statements. The state-changing statements are attached to the arrows in the figure. In order to check whether the SUT has reached a target state or whether it has waited long enough, statements are listed within the state.

In addition to states and transitions, a real test case also includes test steps and test bench-specific functions [25]. The minimalistic test case already contains different types of statements that perform different tasks. Figure 6 shows the mapping of the basic structure of a minimalist test case to the statement types of the test script languages.

It can be seen here that statements of the stimulus type are used to achieve the initial state, the error state and to reset the environment. The pausing of test case execution is needed to wait for the reaction of the SUT. For that purpose, statements of the type runtime behaviour are used. Finally, the system's response is evaluated using Verdicts type statements.

The statement types listed above { Statements for stimulation, Statements for controlling the timing, Statements for verdicts } are necessary to provide a test case's basic functionality. They are available in all test script languages. The most common languages are for testing mechatronic systems (such as cars and aeroplanes): XIL, TTCN-3, CCDL, FLATSCRIPT, C or RTT, and Python. The following differences between these languages are presented in detail to describe the problem statement.

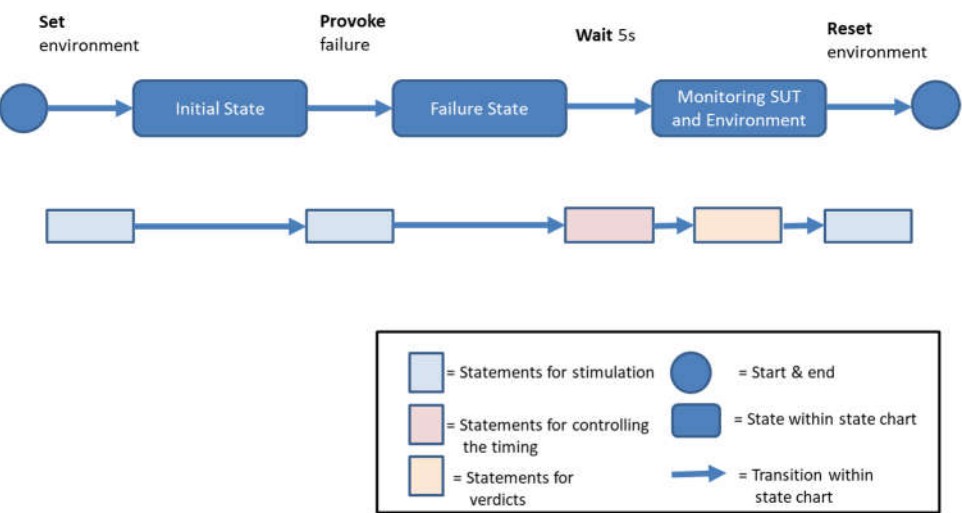

**Figure 6.** Mapping of test case steps to statement types.

### 2.4.2. Structure of Test Script Languages

The available test script languages are imperative programming languages and can be divided into the groups "safe" and "unsafe" [24]. A "safe" language prevents the language from being misused. For this purpose, additional mechanisms such as type security of the variables or unique access methods for data structures are added to avoid inconsistency in the internal representation of the program execution. An example of such an inconsistency would be when the beginning of two arrays in memory have overlapping memory addresses and affect each other. In contrast, "unsafe" languages do not have these restrictions. The programs can be checked statically as well as dynamically at runtime. An overview of common programming languages and the test script languages listed above are listed in Table 1.

**Table 1.** Classification of programming languages.

|  | **Statically Checked** | **Dynamically Checked** |
|---|---|---|
| **Safe** | ML, Haskell, Java, CCDL, FLATSCRIPT, XIL | Lisp, Scheme, Perl, Postscript |
| **Unsafe** | C, C++ | Python, RTT |

Apart from the two groups of test script languages, the typing, the programming paradigm, and the execution modus have an essential impact on the syntax of a test script language. Examples of the impact of the programming paradigm and type safety are shown for CCDL, JAVA and Python in Figure 7.

All three examples set the signal z to the value 5, while the test cases' syntax differs. The interoperability of two test cases from different test script languages must overcome the presented heterogeneity related to typing, programming paradigm, and execution modules. The benefit would be to gain two test cases executed on the supplier's test bench or in any supply chain's test processes.

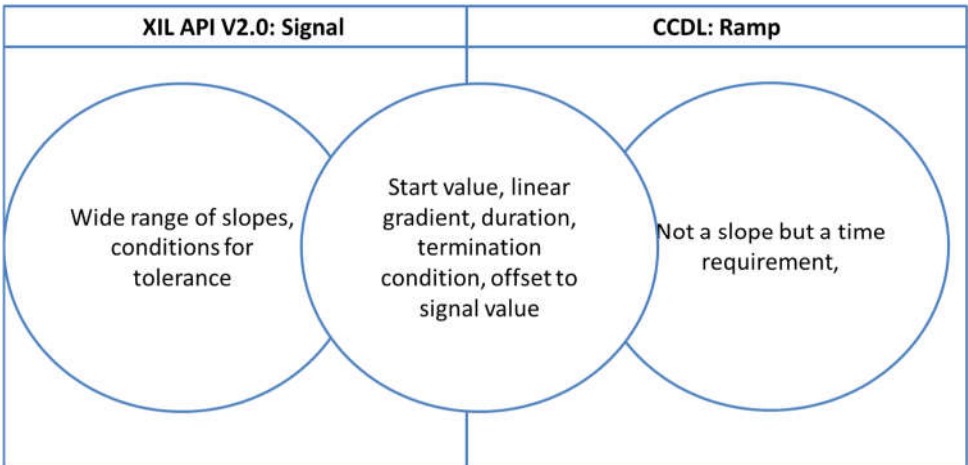

**Figure 7.** A similar test case in different test script languages.

2.4.3. Heterogeneity in Language Complexity of Test Script Languages

The different test script languages are heterogeneous according to their structure, which is reflected in their syntax. Another heterogeneity makes it difficult to integrate test cases from the integrator on the supplier's test bench, namely the heterogeneity in the test process-specific statements. A test process-specific statement is a language feature that is necessary to stimulate and check the behaviour of the SUT. The necessary set of test process-specific functions depends on the test criteria, e.g., test goal, test level, and the to-be-validated requirements. These test criteria vary in the test processes within a supply chain, as explained in 2.1. Thus, different properties of test process-specific statements are necessary. In the following, the heterogeneity is shown for the example: ramp statement, which is an adapted version of a loop statement to stimulate a signal.

The heterogeneity of the function ramp for the test script languages CCDL and XIL API (https://www.asam.net/standards/detail/xil/older/ accessed on 15 August 2022) is shown in Figure 8 below. These two test script languages were selected as examples because CCDL is primarily used in aviation, and XIL API is primarily used in automotive.

**Figure 8.** Heterogeneity of the ramp function in the test script languages XIL and CCDL.

The first difference is the name of the statement. While the ramp statement is called signal in XIL API, it is called ramp in CCDL. Different names for similar test process-specific statements make it difficult to compare test cases from different languages without being an expert in each test script language. Apart from the name of the function, the implementation in XIL and CCDL offers a similar subset of the functionality. Apart from it, the language offers domain-specific features. In the case of CCDL, the ramp can be implemented without any predefined slope. Without further transformations, this ramp variant cannot be mapped to the XIL signal statement.

The example of a ramp statement, which is transferable to all other test process-specific statements, shall demonstrate the heterogeneity. This heterogeneity needs to be harmonised to gain applicable interoperable test cases.

### 2.4.4. Interoperability of Test Cases Using Ontologies

Ontologies are commonly used to model information independently from a specific data format and tools. As discussed in Artificial Intelligence (AI), ontologies are formal, partial specifications of an agreement over the description of a domain [26]. It consists of concepts and relations and their definitions, properties and constraints expressed by axioms [27]. Ontologies have been applied as part of the Semantic Web to describe uncountable domains [28]. With this, RDF/XML and OWL ontologies are the formal proceedings to formalise domain knowledge. Herby, ontologies exist on different levels, each with a specific aim. There are aims to cover basic concepts and properties in so-called foundation/upper ontologies. Mascardi mentioned, "Upper ontologies are quickly becoming a key technology for integrating heterogeneous knowledge from different sources" [29]. Examples of upper ontologies include Basic Formal Ontology (BFO), Business Objects Reference Ontology (BORO), Conceptual Reference Model (CIDOC), and Descriptive Ontology for Linguistic or Cognitive Engineering (DOLCE). Domain ontologies use upper ontologies for fundamental concepts and properties but describe all specifics with their own concepts. The integration of test cases into an upper ontology could be realised, for example, with the concept diagrammatic-object of DOLCE.

The design of test cases as ontology is the scope of a domain ontology. A corresponding domain ontology must represent the statements types (see Figure 6) and the overall test case structure. The ontology languages OWL Web Ontology Language [30], as OWL Lite, provides the required language features to represent a test case. The available domain ontologies for testing are, e.g., STOWS [31], SWTO [32] or OntoTest [33] and current improvements [34] focus on the structure and the processing of a test process in the domain of software testing. In doing so, concepts such as test design, test strategy, test goal, requirements, and test execution module are more in focus instead of the content of a test case. These ontologies include test cases as external resources, which can be linked to the ontology. The content of a test case is relevant to the scope of this article, and its reference is not enough. Specific domain ontologies for the domain of HIL testing, which is the scope of this article, are not available. One solution is to extend one of these domain ontologies to represent the overall state structure of a test case and to integrate test automation specific content. The intended usage of the ontology is to use it as an intermediate representation without the need for querying and reasoning capabilities of an ontology. There is no restriction that ontologies must be used for the formalisation of information. Other models in the application domain of model-based testing and software specification are closer to the formal specification of the test case content and could take the role of an intermediate representation. In consequence, the potential reuse of these models is presented in the following.

### 2.4.5. Interoperability of Test Cases Using Test Models

The interoperability of test cases is already possible through different approaches, which are presented in the following.

One approach is to use test models and generate test cases based on test model in one specific test script language [35,36]. In doing so, the test model defines all possible test cases as a state chart, an activity diagram or something similar to a state chart. Corresponding solutions have been developed as commercial products by, e.g., dSPACE GmbH, Verified Systems International GmbH, and Vector Informatik GmbH as the leading test bench supplier in the automotive and avionics sector. The current solutions support the generation only in their test bench specific test script languages. To gain a test case in another "foreign" test script language, the transformation could translate the source test model into a target test model. Subsequently, the test cases could be generated in the target test script language,

and a suitable test case could be selected. There are available generic models to model transformation solutions, such as OPM modelling tool [37] or transformation languages, such as Query View Transformation (MOF QVT). The application of this approach has two fallbacks. The first is the proper selection of test cases within a corpus of test cases, and the second is the suitable configuration to generate the same set of similar test cases. To simplify this approach, the transformation from a test case into a test model would simplify the configuration and selection of the suitable test case. This feature is not provided by the current solutions. Thus, this solution is not applicable to the interoperability of test cases in the supply chain.

The best practice of model-based testing is that test cases can be represented as a subset of a state chart. This research result enables the right selection of the interoperability model in the article's proposal.

The direct transformation of a test case from one test script language into another test script language is also possible. This type of tool is called a transcompiler (source-to-source compiler) and is primarily used for the interoperability of source code. There are two approaches, namely, the rule-based approaches and the application of AI-based approaches. Both approaches have been researched and evaluated. The outcome is successful translation tools. For example, the tool Moses [38] has been applied to translate from Java to C# or from Python 2 to Python 3. [39]. Hereby, "Unfortunately, the resulting translations often lack readability, fail to respect the target language conventions, and require manual modifications in order to work properly." [39]. Ref. [39] proposes an unsupervised translation method in which a sequence-to-sequence (seq2seq) is used to map similar functionality between the languages using a single model. The approach works well in translating source code between programming languages that share the same programming paradigm and can be trained on enormous data (2.8 million open-source GitHub repositories [39]).

In the scope of this application scenario, a translation is needed in which the programming languages are so heterogeneous that they use different programming paradigms. The translation between different test script languages which have different programming paradigms, means that a one-step translation mechanism cannot work because the needed steps vary between the selected languages. For example, the declaration of a single global variable could be made in one step inside an imperative programming language, while the creation of a global variable inside an object-oriented programming language needs the transformation at multiple locations within the target source code and needs to use a stack. Moreover, the training data in the application domain of avionics contains only a couple of test cases for a specific test script language. There is a need to support more than six test script languages in the supply chain for the target domain. Thus, the available solutions for transcompilers cannot be applied. The approach beyond transcompilers, primarily to establish a single model for all test script languages [39], sounds promising and is completely aligned with the baseline to use state charts as test models.

### 2.4.6. Summary

A mechatronic product, such as an aeroplane, is developed using a supplier pyramid. There are various test processes in this supplier pyramid. These test processes implement test cases according to similar requirements but use different test script languages. Once a test process fails, two test processes must work together. For this purpose, the test case must be interchangeable and integrateable. However, the integration is not possible because the test case was implemented in a different test script language. The test script languages differ so much in this application domain (programming paradigm, safety/non-safety) that no transcompiler is available. Thus, the test cases cannot be integrated without additional help. On the other hand, the execution of the failed test case is necessary to find errors in the test processes within the supply chain.

## 3. Proposal of Interoperable Test Cases in the Product Development

The proposal aims to enable the exchange and execution of test cases in other test processes. For that purpose, the approach focuses on harmonising test cases related to their content and executability. Figure 9 presents three steps of the approach, which are presented in detail.

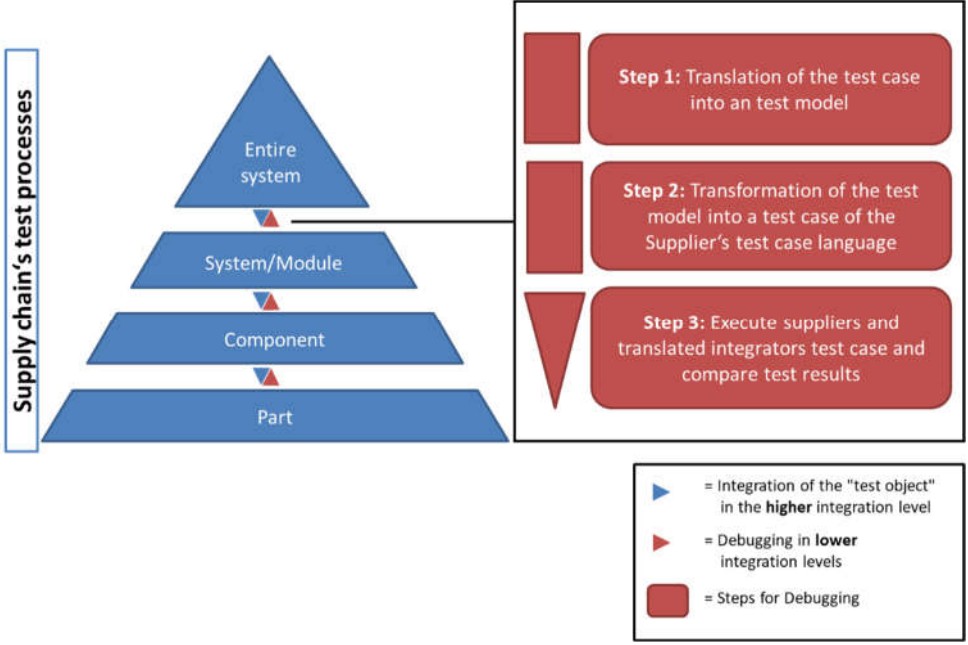

**Figure 9.** Interoperability approach for test cases.

The first step is to translate the integrator's test case from his test script language into a test model. Hereby, the test model represents the test case's logical structure and contains the statements' sequences.

The test case is translated from the test model into the supplier's test script language in the second step. The translation enables the comparison of the integrator's test case and the supplier's test case because both test cases are represented in the same test script language. Thus, static analysis is possible in the second step. To identify all potential differences, the execution of the test cases is necessary.

In the third step, both test cases are executed on the supplier's test bench. Each of the test case executions generates a test report containing signal curves and the outcome of the verdicts. These reports contain all the necessary information to identify the differences between the test cases related to stimulation, verdicts and timing.

The first and second step requires a test model. The test model's specification and implementation are in this paper's focus and will be presented in detail.

### 3.1. Specification of the Test Model

The test model is an interoperability model. It is based on state charts and supports test case specific statements. This interoperable representation shall be used to integrate the test case into the target test process seamlessly.

Each test case is representable as a state chart considering the sequence of statements, which would be sufficient for the test case execution. The paper's goal is to achieve a state chart representation which includes the overall test case structure. For example, compound states for test steps are needed to be visible inside the state charts. Only the Harel state chart is capable of defining compound states. (Franke et al. 2019). Therefore, existing standards for Harel state charts could represent a test case if the expressiveness is high enough to represent all test case related features. For that purpose, an appropriate

interoperability model, such as the Harel state chart, must cover the already defined test script language-specific criteria. An excerpt of the criteria is given in Table 1.

The chosen approach emphasises the usage of SCXML [23] as an interoperability model's data format. It is a standard by the World Wide Web Consortium (W3C) for representing Harel statecharts in an XML-based data format. The integration of Harel state charts enables the representation of states, compound states, and parallel states transparently. Moreover, it supports the extensibility of defining the necessary boolean expression language, location expression language, value expression language and scripting language. The mapping of needed test case related expressiveness, which covers the test case criteria groups (Criteria for data types, Criteria for transitions, Criteria for global conditions, Test specific functions), is shown in Table 2 and has been derived in the research project STEVE [21].

**Table 2.** Coverage of needed test case related features with SCXML features.

| Criterion | Covered by SCXML | Have to be Covered by the Extension |
|---|---|---|
| **Criteria for data types** | | |
| Encapsulation | | X |
| Visibility | | X |
| Primitive data types | | X |
| Complex data types | | X |
| Ranges data types | | X |
| **Criteria for transitions** | | |
| Arithmetical terms | | X |
| Temporal conditions | X | |
| Event based conditions | X | |
| Boolean condition | | X |
| Sequences | X | |
| Parallel sequences | X | |
| Quantifier | X | |
| **Criteria for global conditions** | | |
| Encapsulation | | X |
| Temporal conditions | X | |
| Invariant condition | X | |
| Cyclic condition | X | |
| Predicate terms | | X |
| **Test specific functions** | | |
| Requirement mapping | | X |
| Logging | X | |
| Error injections | | X |

Table 2 demonstrates in the column Covered by SCXML that the expressiveness of SCXML is insufficient to represent a test case. The extension mechanism shall add the missing language features. The column "Have to be covered by extension" summarises the added language feature. The following section describes the specific extension for each of them in detail.

**Data Types**

The proposed extension requires the possibility to represent primitive as well as complex data types. Moreover, encapsulation is required to represent object-oriented data. For that purpose, each data (<data>) shall be represented as a JSON string. For that purpose, the data attribute expr is used. Each JSON String contains a couple of attributes describing the data as well as its content. Each JSON value represents the value whereby it can be a value for a primitive data type or a JSON String. Thus, it supports complex as well as encapsulated data. The remaining JSON attributes are applied to describe the data type on its own. Mandatory information is the definition of whether it is a primitive or a complex data type. The applied data type, the unit and its original name are mandatory attributes supporting the right interpretation.

**Arithmetical Terms**

The extension regarding arithmetical terms and Boolean conditions depends on the chosen language. Appropriate languages such as General Expression language or ASCII Math are applicable. In both cases, an application domain has restrictions on supported language features. For example, the definition of an integral is not possible if the test case shall meet the hard real-time restriction. Similar examples are present for other application domains.

**Test specific Functions**

The extension concerning test-specific functions needs the integration of the requirement mapping and the error injections. The requirement mapping aligns specific stimuli and checks to requirements. This alignment is necessary to support automated evaluation and report generation as part of test automation. Therefore, the JSON strings of all set commands (<assign>) and all checks (<if> and <log>) are allowed to carry a requirement id as part of the data definition. The support of error injections and other test bench-specific methods are representable via predefined invokes (). Not all test-specific functions are accessible via . A subset of common test-specific functions have been selected in the research projects STEVE [40] & AGILE-VT [41] to be supported as a natively implemented set. This set is called TASCXML and has been worked out by Airbus and the leading test bench suppliers dSPACE, Vector and TechSAT. Table 3 summarises the current TASCXML statements in version 5.1 of Generic SCXML.

**Table 3.** TASCXML statements.

| ID | Tascxml Statement | Description |
|----|-------------------|-------------|
| 1 | <tascxml:set> | Sets a signal to a specific value |
| 2 | <tascxml:get> | Reads the current value of a signal ls |
| 3 | <tascxml:result> | Logs a result (verdict) of the test case |
| 4 | <tascxml:ramp> | Triggers a ramp that changes a signal for a specified duration |
| 5 | <tascxml:sine> | Generates a curve shape of a sine curve |
| 6 | <tascxml:sawtooth> | Creates a waveform of a sawtooth |
| 7 | <tascxml:pulse> | Generates a waveform of a pulse |
| 8 | <tascxml:verifytolerance> | Verifies whether the value of a signal is within a certain tolerance range |

Thus, the required extension of SCXML is possible via available technologies, such as JSON, GES or ASCII Math and developed extensions. The basic structure of a test case in SCXML needs to be specified to achieve exchangeable test cases. The research project STEVE & AGILE-VT has worked out best practices for that purpose. The current version (version 5.2 of Generic SCXML) predefines the states StartUp, Logic and TearDown as mandatory states for test objects. A test object (TA) can be a test case or a monitor. A basic structure for a test case is shown in Figure 10.

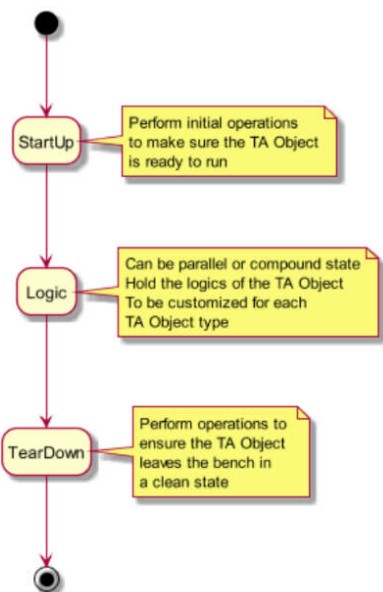

**Figure 10.** The basic structure of a test case in SCXML.

Test cases in CCDL, Python, and RTT define test steps to structure the logical test proceeding. This feature is needed and is added normally inside the test reports. For that purpose, SCXML was extended to a set of predefined events to add meta-information, such as test steps. An overview of the currently supported meta-information as events are shown in Figure 11.

| ID | Payload in JSON |
|---|---|
| testcase.enter.teststep.* | {"label":"<label>","stateid": "<stateid>","teststepdescription": "<teststepdescription>","timestamp": "<timestamp>"} |
| testcase.enter.state.* | {"stateid": "<stateid>","timestamp": "<timestamp>"} |
| testcase.enter.{startup,teardown}.* | {"timestamp":"<timestamp>"} |
| testcase.done.{startup,teardown}.* | {"timestamp":"<timestamp>"} |
| testcase.done.{state,teststep}.* | {"stateid": "<stateid>","timestamp": "<timestamp>"} |

**Figure 11.** Predefined events to define meta-information of a test case. The stars are wildcards and enable the customization of an event for a concrete Test case.

In summary, Generic SCXML is capable of modelling a test case as a state chart whereby the basic structure and additional meta-information for logical structures are predefined. The common feature set implements the functionality to stimulate a signal to check the behaviour and the timing provided by TASCXML.

### 3.2. Transformation between Test Cases and Test Model

The translation of a test case is based on the common steps of syntactic and subsequent semantic processing in compiler construction. While the syntactic processing is based on parse trees, the semantic processing was specifically processed in the direction of a state-based perspective. The translation process from a test case to an SCXML model is shown in Figure 12.

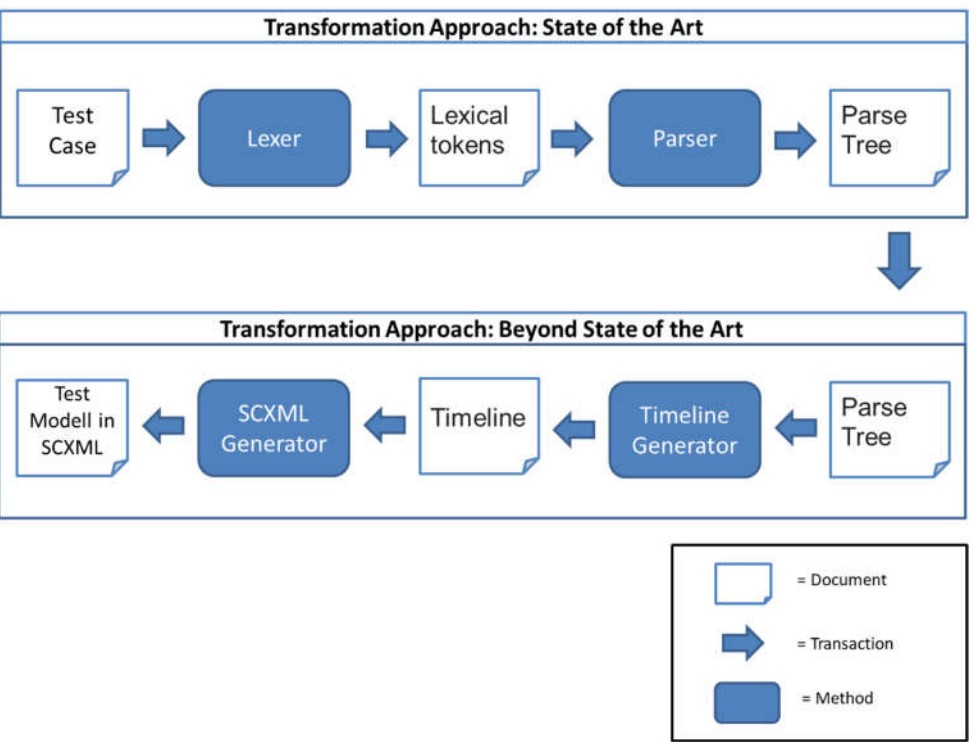

**Figure 12.** Transformation approach based on source-to-source compiling.

For each test script language, a specific lexer and parser are required. The definition and creation of such tools are state-of-the-art and work similarly for all test script languages. The result of the execution of a parser is a parse tree, which models the test case structure as a tree. This tree depends on the properties of the case language according to, e.g., the programming pattern or the type safety.

The harmonisation step to achieve interoperability is located in the timeline generator. The timeline generator is test script language-specific and generates from the parse tree a sequence of statements. Hereby, the statement sequence is a timeline that defines when each statement should be triggered. The timeline does not define how long the statement runs, but rather, it defines whether the statement is blocking or non-blocking. Based on the timeline, the SCXML Generator first creates the overall test case structure (StartUp, Logic, TearDown) and subsequently adds all statements chronologically. In doing so, blocking statements are added as actions inside the predefined states. Parallel statements can be added by creating a parallel state which would be the first child of the state Logic.

The translation from a test model into a test script language works similarly. For that purpose, a test script language-specific generator is needed, which first generates the test case's overall structure and subsequently adds all statements chronologically based on the SCXML states and an SCXML action to statement mapping.

## 4. Application Scenario

The application scenario focuses on exchanging a test case from the integrator to the supplier. In the scenario, the SUT is the high lift system of an aircraft. In the following, the high-lift system is presented as a mechatronic system. Subsequently, the evaluation of the proposed solution is described in detail.

### 4.1. The System under Test

Figure 13 presents an example of an aircraft's basic structure of a mechatronic system. It demonstrates the simplified movement of a slat.

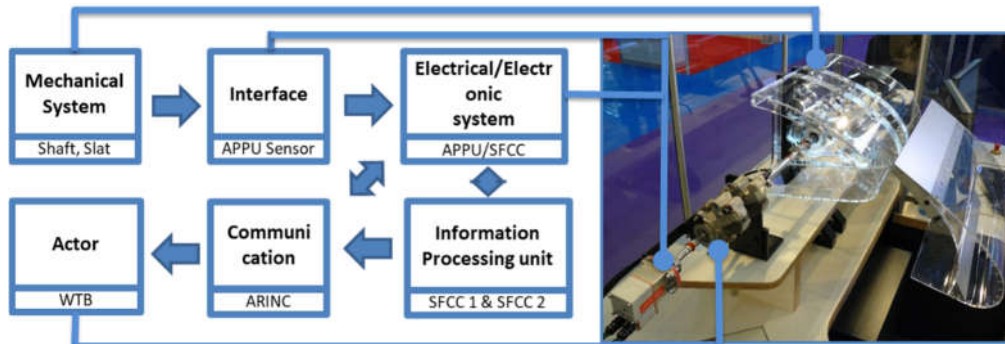

**Figure 13.** Application scenario high-lift of an aircraft.

The slat system is a high-lift device that is usually mounted on the wing. The slat's goal is to adapt the wing's angle of attack according to the flight phase. The slat can be extended or retracted by an electrical motor for that purpose. A computer controls his movement, which is presented in the following.

As shown Figure 13, the Slat/Flap Control Computer (SFCC) controls the shaft movement while the Asymmetric Position Pick-off Unit (APPU) 's sensor sends back the current angle at the tip of the shaft. If the angle reaches the target value, the wing tip brake (WTB) stops the shaft movement, and the slat's extending/retracting stops. The movement happens on both wings in parallel using the same SFCC and has to be monitored and synchronised. One of the possible safety-critical failure cases is asymmetry. As a result of this, an angle difference between the left and right APPU appears. A possible cause is a break in the transmission [42].

The test process is responsible for the assurance of the operational safety of safety-critical systems. For that purpose, the development process implements the test process in parallel to the development activities. In doing so, the test process must prove the mechatronic product's operational reliability and safety. It includes the mechatronic product and its loosely coupled mechatronic systems [43].

From the testing perspective, each integrated mechatronic system, including its parts, is a black box; only interfaces can stimulate the input and provide the output. Moreover, the mechatronic system's correct behaviour can only be checked based on the system under test (SUT) 's reaction. With this, the SUT considers the mechatronic product's properties.

Requirements define which stimuli are appropriate and which behaviour is correct—the complexity of the test process depends on the number of systems and the complexity of their interactions. The application scenario test cases define the stimuli and expected behaviour of the high lift system. The stimulation and the monitoring focus on the SFFCs 1 & 2, because it is the control system of the high lift system. In the following, the exchange of a test case from the integrator to the supplier focusing on the right behaviour of the SFCC 1 & 2 are presented-.

*4.2. Evaluation*

The evaluation aims to demonstrate that test cases can be exchanged and integrated between different test processes in a supply chain. For that purpose, the evaluation focuses on the exchange of a test case by the integrator to his suppliers. In doing so, Airbus takes the role of the integrator, and dSPACE takes the supplier role. The following evaluation scenario does not reflect the real supply chain, but the selected test case is real.

Airbus implements test cases for the high lift in the test script language CCDL. Thus, a test case in CCDL shall be exchanged with the supplier dSPACE. The challenge is that dSPACE does not use CCDL as a test script language and cannot execute CCDL test cases on their test benches. The primary test script language of dSPACE in this scenario is RTT, based on Python.

Executing the integrator's test case at dSPACE is only possible if the test case is translated into RTT and is executable on a dSPACE test bench. For that purpose, the CCDL

test case is translated via SCXML into RTT, the generated test case and a similar RTT test case by dSPACE are executed on the same test bench, and finally, the signal curves of the executed integrator's test case are compared with the signal curves of the dSPACE test case. The specific translation steps are shown in Figure 14.

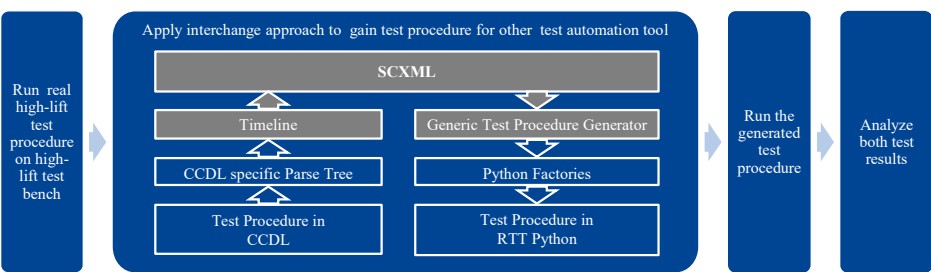

**Figure 14.** Transformation approach between test script languages by Rasche [23].

### 4.2.1. Selected Test Cases for the High Lift System

The test case is designed to check a monitor function in the high lift system. For that purpose, a couple of flap positions and drive sequences are focused. Thus, for each defined position, a drive sequence is initiated via a flaps lever command, and the test evaluates the high lift behaviour according to the requirement-based specification.

To compare the test results, the original test case was executed on a HIL test bench at Airbus that interfaced with an SFCC.

As can be seen in the extract of Figure 15 by [23], the RTT test case (on the right) looks very similar to the selected CCDL test case (left), and both representations are human-readable. However, some information is affected by the translation process. For instance, constants contained in the original scripts are resolved to their numerical representation during the conversion process. They can therefore be displayed only as a number in the resulting representation.

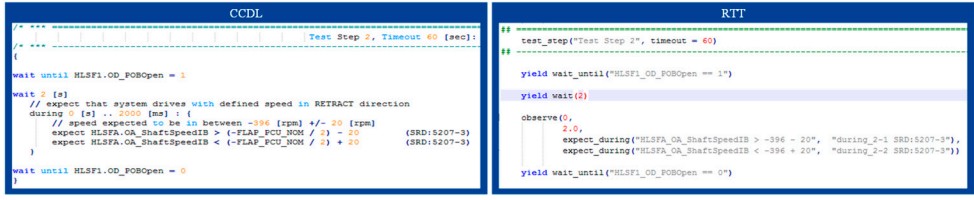

**Figure 15.** Translation result of a test case from CCDL to RTT.

The expected behaviour of the translated CCDL test case and the RTT test case are that they stimulate the SUT the same way, trigger the monitor function of the SFCC at the right time and check the correct behaviour of the SFCC.

### 4.2.2. Evaluation Result

Figures 16 and 17 by [23] show the signal curves that the execution of the RTT test case has created. The signal curves are comparable to the signal curves of the original CCDL test case, which was executed on the Airbus test bench. Thus, the results prove that the resulting test case represented in RTT can be executed on the target system, where it behaves as expected.

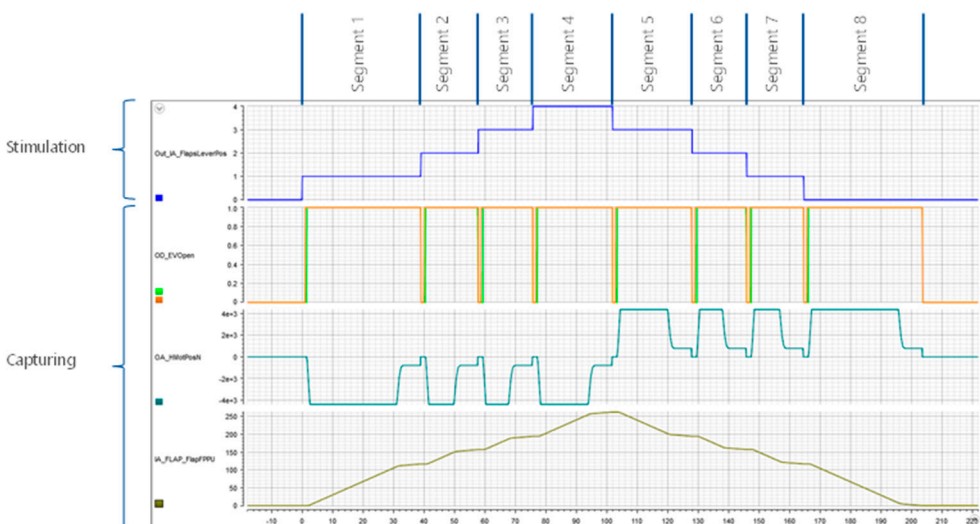

**Figure 16.** Signal curves of the executed RTT test case (B), taken from [23].

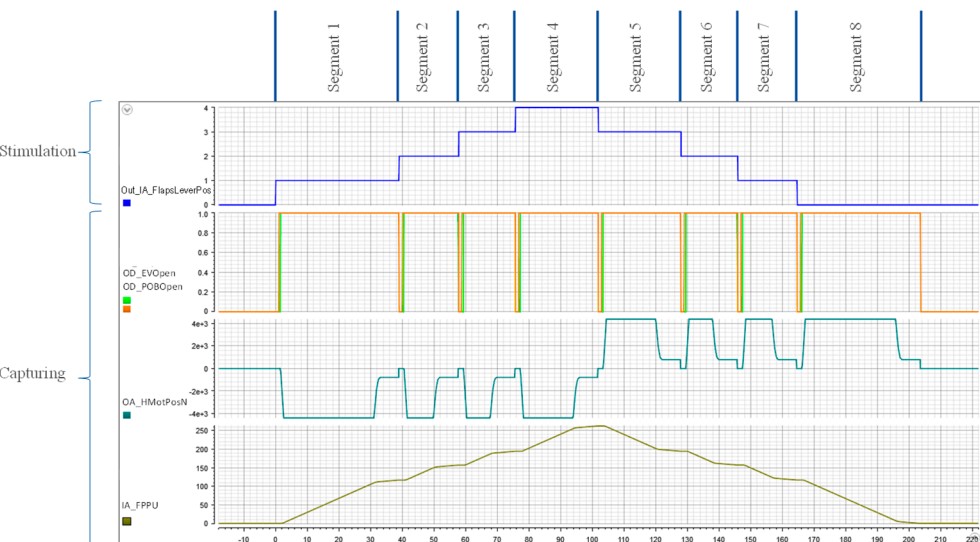

**Figure 17.** Signal curves of the executed RTT test case (A), taken from [23].

The evaluation demonstrated successfully that a test case from the integrator could be translated to the test supplier's test script language and behave comparably. Thus, the interoperability of the test case was established in the supply chain.

## 5. Discussion

### 5.1. Challenges and Findings

The test processes are heterogeneous in a supply chain. The heterogeneity is given because test processes within a supply chain test different systems on different test levels and need to fulfil different system element requirements. The heterogeneity becomes a challenge as soon as the test processes work together. When working together, they have to find the root cause of the faulty behaviour of a test case. For this purpose, test cases serve as a central exchange item. The problem with exchanging test cases is that they are implemented in different test script languages and are not easily interchangeable. One result of the investigation is that exchanging the test case can significantly speed up the search to identify the root cause in the supply chain's test processes. However, this only applies if test cases are automatically translated & integrated into test processes, can be executed, and the test reports can then be compared between the test cases.

In order to ensure interchangeability, the interoperability of test cases must be guaranteed. Interoperability is defined as "the ability of two or more systems or components to exchange and use information" [15]. This article's scope means that test cases can be translated between test script languages. For that purpose, the approach uses a source-to-source compiling in combination with an interoperability model.

The findings show that the test script languages have a similar language scope (feature set) for a given application domain, such as HIL testing. In this way, the language offers similar capabilities for stimulation, monitoring, and timing control, while their syntax can differ completely. This semantic similarity was used to specify a test model as an interoperability model for abstracting test script languages for HIL tests. The result is Generic SCXML. The evaluation pointed out that within the framework of the research projects STEVE and AGILE-VT, test cases from different test departments for different mechatronic systems(doors, oxygen supply, flaps) of an aircraft can be successfully translated into Generic SCXML models. It was also shown that test cases from CCDL and RTT could be translated into Generic SCXML and backwards. This makes the first time for the interoperable exchange of test cases between the test processes within the supply chain possible.

### 5.2. Alignment with Existing Studies

The presented solution focuses on the interoperability of test cases. For that purpose, Generic SCXML is applied as an interoperability model to represent test cases over the boundaries of a specific test script language. Hereby, SCXML is a state chart defined by the W3C. The technology state chart is a common approach to model test cases as test models in the domain of model-based testing [36,44]. Thus, the selection of SCXML is completely aligned with the provided tool landscape of the test bench supplier. Moreover, the execution of test cases in SCXML is also possible. For that purpose, the SCXML standard (https://www.w3.org/TR/scxml/#eventioprocessors accessed on 15 August 2022) defines the baseline for an execution engine. Integrating SCXML test cases into the MBSE & MBT models is also possible because they share the underlying technology state charts. With this, SCXML as an interoperability model can be completely integrated into a test process's design, specification, implementation and execution phase.

Integrating interoperable test cases into the supply chain management fits the evolvement of the supply chain into digital supply chains [9,10]. With this, test cases can be shared as digital resources or used to enhance the digital twin concerning procedural test knowledge. In the latter case, a domain ontology could be specified for test cases based on STOWS [31], SWTO [32] or OntoTest [33] to enable querying and integration capabilities of the Semantic Web. In both cases, the versioning of shared test cases, which are also evolving during a test process, is required and covered by the concept of Digital Threads. Accordingly, the interoperability model, including the source-to-source compiling, can be integrated into the current state of the art for digital supply chains and can expand it based on knowledge management.

### 5.3. Implications for Methodology

The heterogeneity of test script languages concerning type safety, applied programming paradigms and features are a challenge for semantic data integration. From the theoretical point of view, a test case is a program that is written in a specific programming language. Thus the assumption "Translating source code from one Turing-complete language to another is always possible in theory" by [39]. Thus, it proves that a translation is possible. The proof for correctly translating programs between similar programming languages has been listed in [39]. The available tools use transcompilers for that purpose, which enable a source-to-source translation. This article aims to use transcompilers also for test script languages which are not similar. With this, there are two differences between the concept of source-to-source compiling and the presented approach.

The first difference is that we have not developed a transcompiler for each tuple of test script languages. That means we have not developed a transcompiler for example for (CCDL <-> RTT, Flatscript <-> CCDL, etc.). The reason is that the maintenance effort for future integration of new test script languages or new versions of already supported test script languages would be too high. Instead of developing a 1:1 transcompiler for each language, we develop for each new language two transcompilers. The first transcompiler translates from the test script language into SCXML, and the second transcompiler translates from SCXML to the test script language (see Section 3.2). This proceeding guarantees a constant effort to integrate new test script languages. For that purpose, these compilers have an encoder and a decoder for the specific languages, as well as the usage of SCXML as a single model for the representation of the test case content. The article's solution components are entirely aligned with this approach, whereby the decoder is the combination of parse tree/timeline, and the encoder is the SCXML Generator.

The second difference is that the translation of a test case between test script languages does not cover the complete semantics. The part required by the test script language's structure is ignored. That means the decoder reduces the information content. Thus, this solution does not perform a 1:1 translation. Instead, this solution aims to translate the functional point of view of the test case but not the necessary syntax and syntactic sugar. For that purpose, a subset of the statements focusing on the stimulation of a SUT, checking the SUT state and enabling the classification of a test run in passed/failed was in focus. However, the reduction is no problem because the test script specific syntax is added by the encoder if the test case is translated back for a specific test case language.

*5.4. Implications for Practice*

The aim was to improve the collaboration between test processes within the supply chain. The improvement addresses the failure search if an integrator's test case fails and the supplier needs to identify the root cause for its faulty product. Up to now, the interoperability of test cases within the supply chain is not given, and seamless integration of test cases at different suppliers is impossible. Thus, integrating a 'foreign' test case into the supplier's test process is a manual task and needs an expert in the test script language of the integrator and the supplier's test script language. This expertise is necessary because the supplier's test process needs to understand how the integrator has tested its product. For that purpose, he needs to understand how its product and the overall SUT have been stimulated and what was defined as the correct behaviour. In addition to the static analysis, the integrator's test cases need to be executed on the supplier's test bench to identify potential delays or temporal shifts in the stimulation or the SUT reaction. For this purpose, the test case had to be manually translated into the supplier's test language, executed and compared with the supplier's test results, which caused a high manual effort performed by experts. The need to replace these manual efforts with an automated approach depends on the frequency of how often test case needs to be exchanged between suppliers. As shown in the introduction, aircraft design is becoming more and more complicated [6], and the test cases in the test processes are also becoming more complex and, therefore, prone to errors. Thus, the implication of a potential test automation improvement is very high in practice.

Concerning applying the proposed solution in practice, the proposed solution achieves the interoperability of test cases in the test processes within the supply chain. It means that a test case in a test script language A can be translated into a test script language B without manual effort and needing an expert in test script languages. Thus, the failure search as part of the collaboration between test processes is more straightforward because the exchange of test cases, the translation of the test case and the subsequent execution and test report comparison can be automated. In the scope of this article, the step for test case translation has been solved. The other steps are straightforward in their implementation because the available capabilities of the test management tools can already provide them. Translational capabilities can be provided everywhere where a complex mechatronic product is developed within a supply chain. Accordingly, the test case translation can be

used in practice in the development of vehicles, trains, and aircraft, because they rely on hardware in the loop tests. The interoperability of test cases enables the aggregation of test cases to procedural test knowledge, enabling a wide range of information-driven services. Thus, this approach is an enabler for upcoming research activities in the field of knowledge management for test processes.

*5.5. Limitations and Future Research*

The proposed approach enables the interoperability of test cases for HIL tests of mechatronic systems. Thus, test cases focusing on stimulating the SUT via signals and subsequent checking of the correct behaviour are supported. Hereby, there are no restrictions according to type {safety, non-safety} and the selected programming paradigm of the test script language. In theory, each available and future test script language can be supported. From the practical side, the proposed solution has a test script language specific decoder and encoder, which implements the translation from a parse tree to a timeline and the translation from an SCXML model to a specific test script language. That means, for each new supported language, these language-specific software modules must be implemented. Integrating new languages does not require further research activities, but software development projects can do the integration straightforwardly. The current limitation of the prototype is the support for test script languages CCDL and Python, whereby Python has been specialised as RTT by test bench specific libraries. The presented approach and prototype do not reflect the concrete provision of translated test cases within the supply chain. For that purpose, new digital processes, rights & roles concepts, copyrights and other considerations are needed to be worked out. These research questions are related to digital supply chains and are under development [9,10,12].

Translating test cases from different test script languages into the same interoperability model enables the creation of a knowledge base fed by test cases. This knowledge base could replace the heterogeneous silos of test cases spread over the supply chain test processes. Creating such a knowledge base would collect the implicit knowledge within the test case of how a specific system can be stimulated and checked from different perspectives. Each test process in the supply chain has its view of the system and test goals; therefore, it has its own variant of test cases. Aggregating all test cases into one interoperability model could create a harmonised view of an aircraft as a system under test. Subsequently, information-driven services could be provided to the test processes. For example, a test engineer could ask for a given signal and how he could stimulate this signal considering its surrounding from the knowledge base, or a test engineer could ask the knowledge base how to check a specific signal. The creation of this knowledge base and the subsequent development of information-driven services are planned as future research. The knowledge base is also applicable to other phases of the product lifecycle. In the middle of life (MOL) scope, the provision of test knowledge could also improve predictive maintenance activities [44]. The research field of digital twins could also benefit from the integration of test knowledge to determine what is expected and abnormal behaviour. The first steps in exploiting design knowledge are already evident for digital twins of aircraft [45].

## 6. Conclusions

The initial situation was that exchanging test cases between the test processes in the supply chain was impossible. This is because the test cases are implemented in different test script languages. As a result, an exchanged test case cannot be integrated into the test process and used for the failure search without additional effort. In order to make this possible, the exchanged test case previously had to be implemented again in the target language. The presented approach enables the automatic translation of test cases between the test script languages. As a result of this, the approach is generic and can be adapted for any test script language. The evaluation showed that test cases could be translated from CCDL to RTT and showed similar runtime behaviour. The only differences resulted from the test bench properties. The presented approach enables the aggregation of procedural

test knowledge across the boundaries of test script languages and creates a knowledge base for test cases for the first time.

**Author Contributions:** Conceptualisation, M.F. and K.-D.T.; writing—original draft preparation, M.F.; writing—review and editing, K.-D.T.; supervision, K.-D.T. All authors have read and agreed to the published version of the manuscript.

**Funding:** This research has been funded by the Federal Ministry of Economics Affairs and Climate Action of Germany (BMWK) in the project STEVE (project number 20Y1301G) and AGILE-VT (project number 20X1730D).

**Institutional Review Board Statement:** Not applicable.

**Informed Consent Statement:** Not applicable.

**Data Availability Statement:** Not applicable.

**Conflicts of Interest:** The authors declare no conflict of interest.

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
