# Peer review of "Interoperable Test Cases to Mediate between Supply Chain’s Test Processes"

_information, doi:10.3390/info13100498_

Round 1
Reviewer 1 Report
I personally found the paper interesting and potentially suitable for a wide audience.
However, I cannot recommend the publication in the present form mostly because the authors completely ignored the body of knowledge related to interoperability from an information science perspective. Indeed, semantic interoperability and the underpinning technology (Semantic Web technology) is not even mentioned as well as common approaches assuming linked data and ontological support.
Additionally, have authors adopted any of the mentioned approaches? How? If not, why?
Author Response
Dear reviewer,
Thank you for the review and for pointing out the possible improvements. I have revised the respective chapters according to your instructions.
In the following, I give a detailed point-by-point response:
- authors completely ignored the body of knowledge related to interoperability from an information science perspective
Response: I have integrated a strict separation of data, information and knowledge (DIK pyramid). - semantic interoperability and the underpinning technology (Semantic Web technology) is not even mentioned as well as common approaches assuming linked data and ontological support.
Response: I have also introduced the applied definition of interoperability in chapter (2.3 Interoperability as an enabler for information exchange). In addition, I examined the suitability of ontologies as an interoperability model in chapter (2.4.4 Interoperability of test cases using ontologies). - Additionally, have authors adopted any of the approaches mentioned? How? If not, why?
Response: I clarified it in the abstract and inside the introduction. It is an extension of a source-to-source compiler and the interoperability model is based on state charts which is the common meta-model for test models.
Reviewer 2 Report
Interoperable test cases to mediate between supply chain's test processes
The article presents the approach in detail. For this purpose, an interoperability model for test cases is presented. Based on the interoperability model, a cross-translation for test cases between test case languages is shown. The developed translation approach is capable of handling test case languages, which are different with respect to type safety and applied programming paradigms. Moreover, the readability of generated test cases is given to a test engineer. It means that the structure of the test case is readable, as well as the labelling of signals to generate helpful test reports for the test process.
l This is an interesting piece of “Interoperable test cases to mediate between supply chain's test processes” work. Please underscore the scientific value added/contributions of your paper in your abstract and introduction and address your debate shortly in the abstract.
l Introduction needs to include: What has been studied Introduction should be clearly stated research questions and targets first. Then answer several questions: Why is the topic important (or why do you study on it)? What are research questions or objectives? What are your contributions? Why is to propose this particular method (This must come from Literature discussion)?
l The references need to update to 2022. For instance, Chunyan Zhu, Xu Guo & Shaohui Zou (2022) Impact of information and communications technology alignment on supply chain performance in the Industry 4.0 era: mediation effect of supply chain integration, Journal of Industrial and Production Engineering, 39:7, 505-520, DOI: 10.1080/21681015.2022.2099472; and Yudi Fernando, Ming-Lang Tseng, Ika Sari Wahyuni-Td, Ana Beatriz Lopes de Sousa Jabbour, Charbel Jose Chiappetta Jabbour & Cyril Foropon (2022) Cyber supply chain risk management and performance in industry 4.0 era: information system security practices in Malaysia, Journal of Industrial and Production Engineering, DOI: 10.1080/21681015.2022.2116495
l The major defect of this study is the debate or Argument is not clear stated in the introduction session.
l I would like to request the author to emphasis on the contributions on practically, methodology or academically in your discussion session.
l Basically, you should enhance your findings, limitations, underscore the scientific value added of your paper, and/or the applicability of your contributions/shortages and future study in this session.
Author Response
Dear reviewer, Thank you for the review and for pointing out the possible improvements. I have revised the respective chapters (abstract, introduction, discussion) according to your instructions. In the following, I give a detailed point-by-point response:l This is an interesting piece of “Interoperable test cases to mediate between supply chain's test processes” work. Please underscore the scientific value added/contributions of your paper in your abstract and introduction and address your debate shortly in the abstract.
Response: I updated the abstract and introduction to underscore the scientific value added/contributions.
l Introduction needs to include: What has been studied Introduction should be clearly stated research questions and targets first. Then answer several questions: Why is the topic important (or why do you study on it)? What are research questions or objectives? What are your contributions? Why is to propose this particular method (This must come from Literature discussion)?
Response: I've edited the first sentence accordingly. In addition, I have reduced the introduction to its core statements. Then I answered the W-questions in the text.
l The references need to update to 2022. For instance, Chunyan Zhu, Xu Guo & Shaohui Zou (2022) Impact of information and communications technology alignment on supply chain performance in the Industry 4.0 era: mediation effect of supply chain integration, Journal of Industrial and Production Engineering, 39:7, 505-520, DOI: 10.1080/21681015.2022.2099472 ; and Yudi Fernando, Ming-Lang Tseng, Ika Sari Wahyuni-Td, Ana Beatriz Lopes de Sousa Jabbour, Charbel Jose Chiappetta Jabbour & Cyril Foropon (2022) Cyber supply chain risk management and performance in industry 4.0 era: information system security practices in Malaysia, Journal of Industrial and Production Engineering, DOI: 10.1080/21681015.2022.2116495
Response: I updated the references where applicable. In this scope, I added the state-of-the-art related to digital twins, digital supply chains and ontologies
l The major defect of this study is the debate or Argument is not clear stated in the introduction session.
Response: I updated the introduction accordingly
l I would like to request the author to emphasis on the contributions on practically, methodology or academically in your discussion session.
Response: I restructured the chapter 5 Discussion and added a new subsection: 5.3 Implication for Methodology
l Basically, you should enhance your findings, limitations, underscore the scientific value added of your paper, and/or the applicability of your contributions/shortages and future study in this session.
Response: I enhanced chapter 5
Round 2
Reviewer 1 Report
The feedback provided has been somehow addressed so I don't have any reason for not recommending the paper for publication.
However, please consider there is still room for improvement, especially in terms of presentation.
Author Response
Dear reviewer,
Thanks again for the review and for pointing out the possible improvements. I have revised the respective chapters
In the following, I give a detailed point-by-point response:
I However, please consider there is still room for improvement, especially in terms of presentation.
Response:
- I improved all figures to be better readable and follow the same layout style
- I improved the figure’s captions to define the figure’s context
- I performed an additional spell check and simplified a couple of sentences
- I restrict the approach’s scope from knowledge to procedural knowledge. Thus, declarative knowledge is not in the scope of this article.
Reviewer 2 Report
accepted
Author Response
Dear reviewer,
Thanks again for the review and for accepting the paper